# Transferable Tactile Transformers for Representation Learning Across Diverse Sensors and Tasks

**Jialiang Zhao**[1]    **Yuxiang Ma**[2]    **Lirui Wang**[2]    **Edward H. Adelson**[1]
MIT CSAIL
[1]{alanzhao,adelson}@csail.mit.edu
[2]{yxma20,liruiw}@mit.edu

**Abstract:** This paper presents T3: Transferable Tactile Transformers, a framework for tactile representation learning that scales across multi-sensors and multi-tasks. T3 is designed to overcome the contemporary issue that camera-based tactile sensing is extremely *heterogeneous*, i.e. sensors are built into different form factors, and existing datasets were collected for disparate tasks. T3 captures the shared latent information across different sensor-task pairings by constructing a shared trunk transformer with sensor-specific encoders and task-specific decoders. The pre-training of T3 utilizes a novel Foundation Tactile (FoTa) dataset, which is aggregated from several open-sourced datasets and it contains over 3 million data points gathered from 13 sensors and 11 tasks. FoTa is the largest and most diverse dataset in tactile sensing to date and it is made publicly available in a unified format. Across various sensors and tasks, experiments show that T3 pre-trained with FoTa achieved zero-shot transferability in certain sensor-task pairings, can be further fine-tuned with small amounts of domain-specific data, and its performance scales with bigger network sizes. T3 is also effective as a tactile encoder for long horizon contact-rich manipulation. Results from sub-millimeter multi-pin electronics insertion tasks show that T3 achieved a task success rate 25% higher than that of policies trained with tactile encoders trained from scratch, or 53% higher than without tactile sensing. Data, code, and model checkpoints are open-sourced at https://t3.alanz.info.

## 1   Introduction

The tactile sensing modality has gained increasing popularity within the robotics community, by providing important fine-grained contact information for dexterous and contact-rich manipulation tasks, such as robot grasping [1, 2] and fabric manipulation [3, 4]. Camera-based tactile sensing [5], a sensing method that operates by embedding a camera beneath a soft elastomer to capture the fine-grained interactions with the environment, is among the most popular methods of tactile sensing for its higher resolution and lower cost. However, camera-based tactile sensors are extremely *heterogeneous*, and there has not been a converged sensor design widely adopted by the robotics community. Different tactile sensors can differ significantly in shapes, types and numbers of cameras, placement and colors of illumination, etc. Such inherent heterogeneity hinders roboticists from building a general-purpose tactile encoder that is transferable across different sensors and downstream tasks.

Existing learning architectures and datasets focus on one specific sensor-task pairing, and when it comes to a newly emerged sensor or task, data recollection and training an encoder from scratch are often required. This issue harms learning efficiency in a more significant way in longer horizon tactile manipulation tasks, where the training of the tactile encoder is guided by a more sparse reward. Intuitively, although different tactile sensors produce vastly different tactile images and various tasks extract different information from a tactile input, there should be shareable latent information due to the inherent similarities in tactile sensing across sensors and tasks. Therefore, it is both technically viable and practically desirable to design an architecture capable of extracting such shared latent representations and transferring them across different sensor-task pairings.

8th Conference on Robot Learning (CoRL 2024), Munich, Germany.

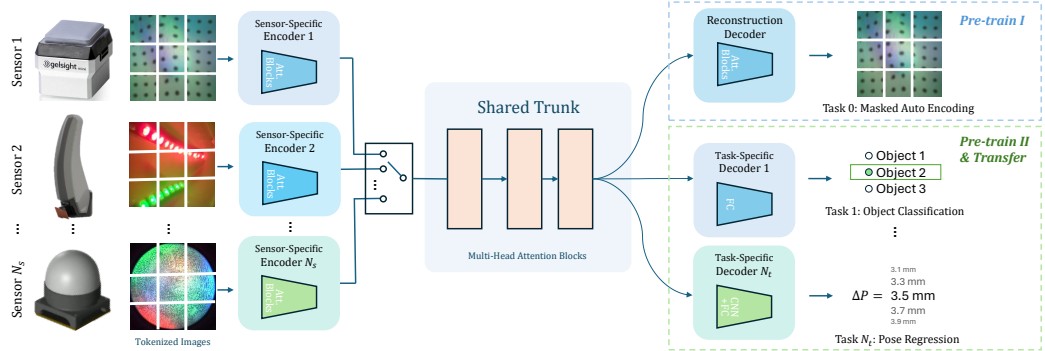

Figure 1: **Architecture illustration of Transferable Tactile Transformers (T3).** T3 learns a shared representation across heterogeneous tactile sensors and downstream tasks with a shared trunk between sensor-specific encoders and task-specific decoders. The encoders and the shared trunk are constructed with transformer blocks. The decoder architectures are chosen according to the types of the tasks: we use transformer for generative tasks like reconstruction for masked auto encoding, MLP for classification tasks, and CNN + MLP for pose estimation.

Learning from heterogeneous tactile inputs shares many similarities with multi-modal representation learning. Recent works in multi-modal foundation models have demonstrated the potential capability to bridge the gap between different modalities and learning a common latent representation, such as in the perception domain [6, 7, 8] as well as the robotic manipulation domain [9, 10]. However, learning such foundation models often requires temporally aligned multi-modal inputs or a distance function that describes the resemblance between inputs from different modalities. In *heterogeneous tactile learning*, the hardware-level disparity often makes it ill-defined to align images collected from different embodiments.

In this work, we tackle the problem of heterogeneous tactile learning using unaligned tactile data, with the goal of learning a scalable representation that can be shared across different sensors and tasks. We first aggregate existing open-sourced tactile datasets and assemble the Foundation Tactile (FoTa) dataset, which is the largest and most diverse tactile dataset so far to the best of the authors' knowledge. We then propose Transferable Tactile Transformers (T3) to learn a shared representation across multi-sensors and multi-tasks. T3 is constructed with sensor-specific encoders, a shared trunk, and task-specific decoders. Our experiments show that T3 pre-trained with FoTa achieves reasonable zero-shot transferability, it can be further fine-tuned with small domain-specific datasets, and its performance scales with bigger network sizes. We also demonstrate that T3 can be used as a tactile encoder for longer-horizon manipulation policies with 3 behavior cloning-based sub-millimeter level electronics insertion tasks. In summary, this paper proposes:

**Foundation Tactile (FoTa):** a dataset containing $3,083,452$ tactile images collected with 13 sensors for 11 tasks in a unified format. It is aggregated with several of the largest open-sourced tactile datasets and additional data collected in-house by the authors.

**Transferable Tactile Transformers (T3):** a neural network framework that learns a shared representation across all sensors and tasks in FoTa. Pre-trained weights are provided, and we demonstrate that T3 can be easily fine-tuned to a new task or a new sensor with few fine-tuning data.

## 2 Related Works

**Camera-based tactile sensors** Camera-based tactile sensors are often constructed with a soft elastomer as the contact medium, a reflective layer, a light source to provide illumination, and one or more than one camera to capture the deformation of the elastomer, which therefore provides a detailed view of the contact surface. Specific design choices in camera-based tactile sensors are highly diverse, such as monochrome [11] v.s. colored [12, 13, 14] in illumination, with v.s. without markers [15, 16, 17], and different form factors such as flat [12, 13], dome-shaped [18, 14], finger-shaped [19, 20], compliant [21, 22], multi-linked [23] or even palm-shaped [24]. Such diversity

and disparity have enabled researchers to use those sensors across a spectrum of applications such as in defect inspection [25] and robotic tool-using [9]. Nevertheless, they also add a substantial barrier in leveraging machine learning to encode the information coming from those sensors. A unique sensor-task pairing often requires data re-collection and neural network re-training.

**Datasets for tactile manipulation** Several large-scale tactile sensing datasets have been released in recent years tackling visual-tactile cross-modality reasoning [26, 27, 28], object or material classification [28, 29, 4], longer horizon tasks such as robotic grasping stability analysis [2], or shape reconstruction [30]. Those datasets also cover interactions with a wide variety of objects, including household objects and toys [2, 26, 27, 28, 30], fabrics and clothes [4], natural objects such as wood and gravel [29]. Quantity-wise, several of those datasets contain well over a million images[26, 30], making the total number of available tactile images on par with ImageNet-1K [31].

While quantity is often not an issue when using existing open-sourced tactile datasets, one significant drawback is that all the aforementioned datasets lack diversity in terms of sensors and tasks. The vast majority of tactile datasets were collected entirely with only one specific tactile sensor. With this constraint, even if a sufficiently well-performing tactile encoder can be trained from one dataset, such a pre-trained encoder will not be able to transfer to another sensor. In order to learn a general-purpose representation for tactile sensing like [32, 33] does in the natural image domain, a large and diverse tactile dataset with a wide variety of sensor-task pairings is a prerequisite.

**Representation learning with heterogeneous data** Learning from heterogeneous data domains, such as taking multi-sensory feedback as inputs or predicting outputs for different tasks, has been an active area of research, especially in the robot learning domain. Many existing works combine tactile, vision, and occasionally other modalities such as robot states and sound, by passing each modality through one modality-specific encoder, then concatenating them and passing them through a decoder network to predict the final output [34, 15, 35, 9, 36]. Apart from learning-driven neural networks, it is also common to use analytical tools, such as factor graphs or convex optimization, to combine predictions from multiple domains [37].

Another branch of multi-modal robot learning actively reasons the relationship among heterogeneous domains and cross-links them to learn a shareable latent representation. Extensive research has explored using contrastive learning to join temporally aligned modalities [6, 10, 38]. To connect vision and touch, Chen et al. designed a multi-modal self-attention mechanism [39] to learn a shared representation, while Li et al. proposed a cross-modal Generative Adversarial Network and were able to predict tactile images from visual images and vice versa [26]. Bachmann et al. applied masked auto-encoding (MAE) on aligned RGB images, depth maps, and semantic segmentation and learned a shared representation across three domains [7]. However, although cross-modality representation learning shares a lot of commonalities with heterogeneous tactile representation learning, one significant difference is that heterogeneous tactile representation learning lacks *aligned* data collected with different sensors for different tasks. It is often ill-defined to infer the resemblance of two tactile images collected from different hardware, which is a prerequisite for many self-supervised learning techniques. It is therefore desirable to devise a new architecture that is able to learn from **unaligned** tactile data and produce a representation that is transferable across different tactile sensors and different tasks.

## 3   The `FoTa` Dataset

We provide a large tactile dataset by aggregating existing public datasets as well as adding new data collected in-house. The `FoTa` dataset is provided in a unified, I/O efficient WebDataset [40] format. It is by far the largest and most diverse dataset for tactile perception, with $3,083,452$ data points collected on 13 tactile sensors, and high-quality labels are provided for 11 tasks. Mixture of constituents of `FoTa` is illustrated in Fig. 2.a, and example tactile images from each sensor are shown in Fig. 2.b. Details and statistics about public datasets aggregated in `FoTa` as well as the pre-processing method are provided in Appendix A.2.

Besides aggregating publicly available tactile datasets, we add new data collected with several recently emerged tactile sensors including GelSight Finray [5], GelSight Svelte [19], GelSight Wedge [13], DenseTact 2.0 [14], and GelSight 360 [18]. Those new data were also used for final performance

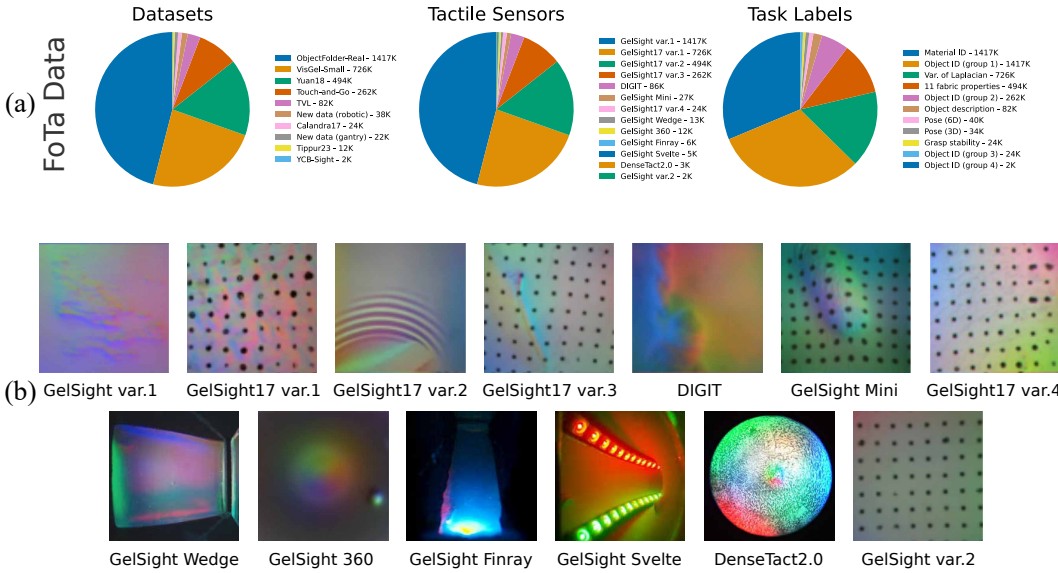

Figure 2: `FoTa` **dataset visualizations.** (a) We show the mixture and distribution of constituent datasets, sensors, and tasks of the `FoTa` dataset. Note that not all tasks are utilized in the training of T3. (b) We visualize one tactile image from each constituent sensor in `FoTa`. Note that in the training of T3, similar sensors share encoders. For example, *GelSight17 var. {1-4}* share the same encoder, and *GelSight var. {1-2}* share the same encoder.

benchmarking as well as transferability testing for the proposed learning framework. Two platforms were built to collect these additional data.

**A 7-DoF robotic platform** collects tactile data by attaching 2 tactile sensors to a parallel jaw gripper of a 7-DoF Franka Emika Panda robot arm and commanding the robot arm to explore an object that is fixed at a known location. Robot pose in $SE(3)$ is recorded for each step and then transformed into the sensor's coordinate frame. Object IDs as well as names are also recorded.

**A 3-DoF gantry platform** is built with a desktop 3-axis CNC mill. Six 3D printed probes with different textures are attached to a force/torque sensor, which is then attached to the Z axis of the CNC. During data collection, a sensor is fixed at a known location, and the CNC machine probes the entire sensing surface of the sensor. Translational poses, probing forces, and probe IDs are recorded for each tactile image frame.

The two systems are introduced in more detail in Appendix A.4.

## 4    Heterogeneous Tactile Learning with `T3`

We propose Transferable Tactile Transformers (T3) as the backbone network to learn from diverse tactile sensors and produce outputs for diverse downstream tasks.

**Network architecture**    A training dataset contains inputs from $N_s$ different sensors and $N_t$ distinguished tasks. T3 is constructed with $N_s$ encoders $\{Enc_1, ..., Enc_{N_s}\}$, one for each individual tactile sensor; $N_t$ decoders $\{Dec_1, ..., Dec_{N_t}\}$, each responsible for one downstream task; and one shared trunk, denoted as $Trunk$. During both training and inference, we ensure that one batch of data always comes from the same sensor-task pairing. For a given pair of data point $(X_i, Y_j)$ that is collected from sensor $i$ for task $j$, the corresponding encoder $Enc_i$ and decoder $Dec_j$ are attached to the trunk. The constituent network components of T3 can take different architectures. In practice, we use ViT [41] in the trunk and encoders for its proven learning capacity and scalability in the natural image domain. For decoders, we use ViT for generative tasks (reconstruction for MAE),

ResNet+MLP for pose estimation tasks, and MLP for classification tasks. More details about network configurations are discussed in Appendix A.5.

**Learning objectives**  For each task $j \in [1, N_t]$, we define a loss function $L_j$. Different tasks require different numbers of tactile images as inputs. For example, tasks like object classification and material classification only require one tactile image to perform, while the pose estimation task requires two tactile images, e.g. $X^1, X^2$, because the prediction goal is the relative pose between the two instances. We pass each individual tactile image through the encoder and trunk separately, and we then concatenate them together before passing them through the decoder. The loss between a data pair $(X_i, Y_j)$ or $([X_i^1, X_i^2], Y_j)$ is calculated as

$$
\begin{aligned}
loss(X_i, Y_j) &= L_j(Y_j, Dec_j(Trunk(Enc_i(X_i)))) \\
loss([X_i^1, X_i^2], Y_j) &= L_j(Y_j, Dec_j(Trunk(Enc_i(X_i^1)) \oplus Trunk(Enc_i(X_i^2))))
\end{aligned}
\tag{1}
$$

where $\oplus$ denotes concatenation.

The training process of T3 is split into two pre-training phases and one optional fine-tuning phase. Pre-training is divided into two phases to improve data utilization and to attain both local fine-grained understanding and global semantic understanding. Phase 1 focuses on more pixel-level understanding using self-supervised learning, and it is able to utilize all data from the FoTa dataset. Phase 2 focuses on semantic understanding and it requires suitable task labels that only a subset of the FoTa dataset possesses.

**Pre-training I: Self-supervised learning with Masked Auto Encoding (MAE)**  MAE was first proposed as a self-supervised pre-training technique for natural images [42] and it has gained increased popularity in other domains such as audio processing [8], point-cloud processing [43], and multi-modal representation learning [7]. It operates by first randomly masking out a certain portion of the original inputs, then it constructs a decoder to reconstruct the missing portion. Neural networks that were pre-trained with MAE have been shown to be able to outperform ones that were trained without MAE in various domains. This pre-training stage is able to utilize all data in FoTa, including unlabeled data (e.g. intermediate tactile images collected during the settling period of a robot grasp in [2]), or data with semantic labels for which loss functions are hard to define (e.g. natural language description of a scene in [27]).

In this pre-training stage, one decoder $Dec_0$ that is designed to reconstruct partially masked images is shared by all domains. $Dec_0$ is constructed with 8 VisionTransformer blocks [41]. The loss $L_0$ is L2 pixel-wise loss normalized within each patch, following [42].

**Pre-training II: Supervised learning with labels distilled from public datasets**  In this stage, T3 is trained under supervision from selected task labels in the FoTa dataset. For each individual task $j \in [1, N_t]$, a decoder $Dec_j$ and a loss function $L_j$ is defined. We define 10 tasks for this training stage.

- Classification tasks include object classification for [2, 29], material classification for [30], and fabric smoothness, fuzziness, textile type classification for [4]. Decoders for each of them are defined as MLP neural networks, and each $L_j$ is a cross-entropy loss function.

- Regression tasks include $SE(3)$ relative pose estimation bewteen two overlapped tactile images for [13, 44, 45] (with ResNet+MLP decoders), and variance of Laplacian (a metric describing the amount of information contained in tactile images, more details in Appendix A.2) estimation for [26] (with MLP decoders). Each loss $L_j$ is a mean squared error loss function.

**Fine-tuning: Supervised learning with task-specific data**  In this phase, T3 is further fine-tuned with data collected for the specific downstream task with the specific sensor that the user aims to use. This stage can be optional if the sensor-task pairing already exists in the pre-training dataset. Architecture-wise, T3 is configured the same way as it is in Pre-training II but with only one target task and sensor.

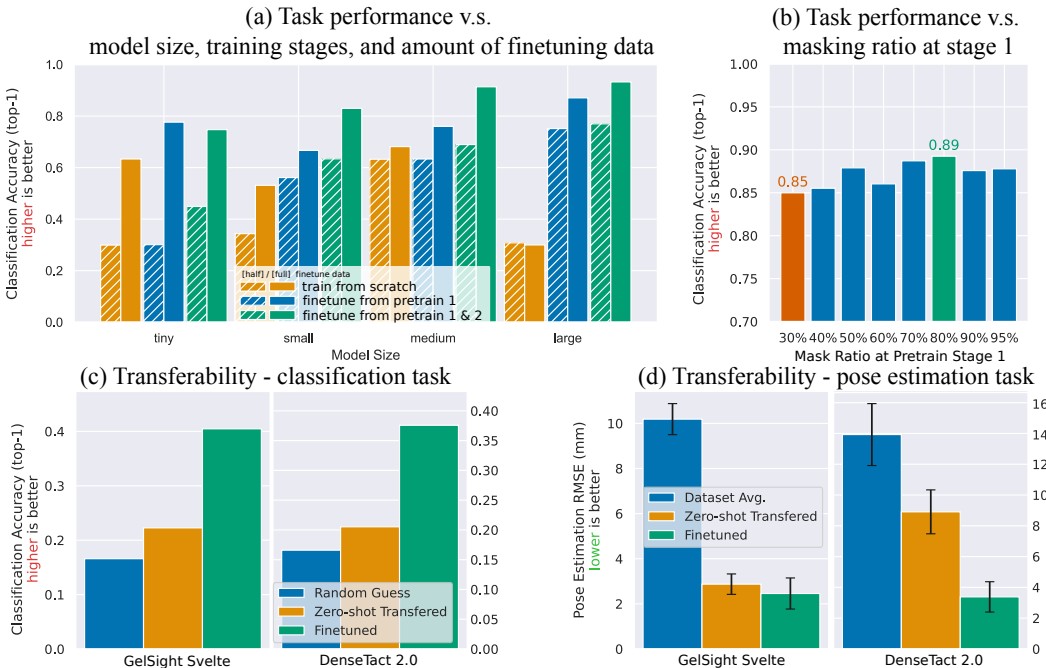

Figure 3: **Experiments on Task Performance and Transferability of T3.** (a) Eval performance with 4 network sizes (`tiny`, `small`, `medium`, `large`), 3 training schemes (train from scratch, fine-tune from pre-train 1, fine-tune from pre-train 2), and 2 amounts of fine-tuning data (half data and full data) (b) Eval performance with different masking ratios during Pre-training I. (c) Transferability test on a classification task. (d) Transferability test on a pose estimation task.

## 5 Experiments and Discussions

In this section we discuss the evaluation performance of T3 with numerous ablations and benchmarks. For both pre-training I and II, we use all data from the `FoTa` dataset except data collected on the gantry system in-house, including the object classification task and the 3D pose estimation task. Those data are therefore not included for pre-training the foundation model and are set aside for evaluation and fine-tuning only. We aim to answer the following questions in this section:

- How to train efficiently and how much does pre-training improve performance? Sec. 5.1.
- Is pre-trained T3 *zero-shot* transferable to new tasks and / or sensors? Sec. 5.2.
- Does `FoTa` provide meaningful improvements in longer-horizon robotic tasks? Sec. 5.3.

### 5.1 The worthiness of pre-training

The fine-tuning and evaluation task for this section is an object classification task (6 categories) for 2 sensors (GelSight Wedge, GelSight Mini) with around 3,300 data points for each sensor. Averaged top-1 classification accuracy from the two sensors is reported.

**The effect of pre-training and the scaling behavior on different network sizes**   We conducted the classification experiment with all combinations of the following: (a) 4 network variations: *tiny*-12M, *small*-45M, *medium*-174M, and *large*-308M; (b) 3 training schemes: train from scratch, fine-tune from pre-training I, and fine-tune from pre-training I & II; (c) 2 amounts of training data: half data (≈1,650 training data) and full data (≈3,300 training data). All 24 experiments were tested on the same validation dataset, which was separated from the training set. The evaluation accuracy is shown in Fig. 3.a. We observed that: (a) Pre-training significantly improved the evaluation performance with all network configurations with an median improvement of 24%. (b) The evaluation performance

improved with larger networks, where *large*'s classification accuracy was 19% higher than that of *tiny*. However, the performance difference between *medium* and *large* was insignificant. (c) When fine-tuned with half of the fine-tuning data, pre-trained networks performed better than trained from scratch networks, and larger pre-trained models performed better than smaller pre-trained models. In fact, for *medium* and *large*, the performances between fine-tuned with half data and fine-tuned with full data were very close, signaling those models generalized better thus they require less data to fine-tune to a novel task.

More qualitatively analysis, including visualizations of the encoder attention maps and the trunk attention maps, can be found in Appendix A.1.

**Masking ratios in Pre-training Stage I**   We tested 8 different masking ratios in the MAE reconstruction pre-training stage, and we performed the same Pre-training II stage as well as the fine-tuning stage on the 8 tests. The evaluation results are reported in Fig. 3.b. We observe a small performance variance, ranging from 85% ($mr = 30\%$) to 89% ($mr = 80\%$), and a masking ratio of 80% yielded the highest performance.

## 5.2   Zero-shot transferability of pre-trained `T3` to new sensors or tasks

We tested to what extent a pre-trained T3 can be transferred to a new sensor or downstream task *with* and *without* fine-tuning.  We picked two novel sensors (GelSight Svelte, DensetTact2.0) and two seen tasks (object classification, pose estimation between two overlapped tactile images). The two seen tasks were pre-trained with data from GelSight Wedge, GelSight Finray, and GelSight Mini. During this experiment, we use the encoder pre-trained for GelSight Wedge as the encoder for GelSight Svelte, and GelSight Mini's encoder in place of DenseTact2.0's encoder.  Results on the 4 combinations are shown in Fig. 3.c and Fig. 3.d. Zero-shot transfer yielded only minor improvements over random guesses in the object classification task; however, it demonstrated significant improvement over dataset averages in the pose estimation task. We further fine-tuned the networks with a small amount of 2,000 data points. The classification accuracy immediately bounced up to 17% with fine-tuning, and the pose estimation root mean squared error (RMSE) was reduced by 5.5mm for DenseTact2.0. On the GelSight Svelte sensor, the pose estimation errors before and after fine-tuning were nearly identical and close to optimal.  These results indicate that zero-transferability can be achieved in certain sensor-task pairings, and that better outcomes can be attained with minimal fine-tuning.

## 5.3   `T3` in long-horizon manipulation tasks

Tactile sensing can be especially helpful in contact-rich manipulation tasks. Numerous human studies have shown that the sense of touch is important for humans to perform delicate manipulation tasks such as surgery [46, 47]. However, most state-of-the-art tactile manipulation research opted to train a tactile encoder from scratch, with fresh data collected with the specific sensor-task pairing, due to the lack of a pre-trained tactile encoder. In the case of long-horizon multi-modal manipulation tasks, training from scratch is often inefficient, due to the sparseness of trajectory rewards and the number of components that need to be trained (e.g. modality encoders and policy networks) especially in many reinforcement learning and behavior cloning applications such as [48, 9]

To investigate whether a pre-trained T3 helps to bridge this gap, we designed a robotic precision insertion task with behavior cloning. The goal of this task is to insert 3 electronics parts: a 3-pin toggle switch, a 12-pin double-stack USB port, and a 17-pin VGA connector, onto a PCB with corresponding mounting holes to each component. This task requires high precision, where the clearance between the holes on the PCB and the pins on the parts is only $0.4mm$. Achieving this precision requires active exploration with tactile feedback. In real-world applications, relying on vision alone is often insufficient due to heavy occlusion.

Our experiment setup consists of two GelSight Wedge tactile sensors mounted on a parallel jaw gripper, which is attached to the end-effector of a 7-DoF Franka Emika Panda robot, a PCB board fixed at a known location on the workbench, the 3 electronics parts, and a RGB camera on the side of the workbench. An illustration of the setup is shown in Fig. 4 a and the parts are shown in Fig. 4.b. Data is collected by controlling the robot in "guide mode", i.e. under zero stiffness control, and a human operator manually grabs and moves the end-effector until the part is fully inserted. At both data collection and inference time, the sockets are fixed, and we add randomizations to

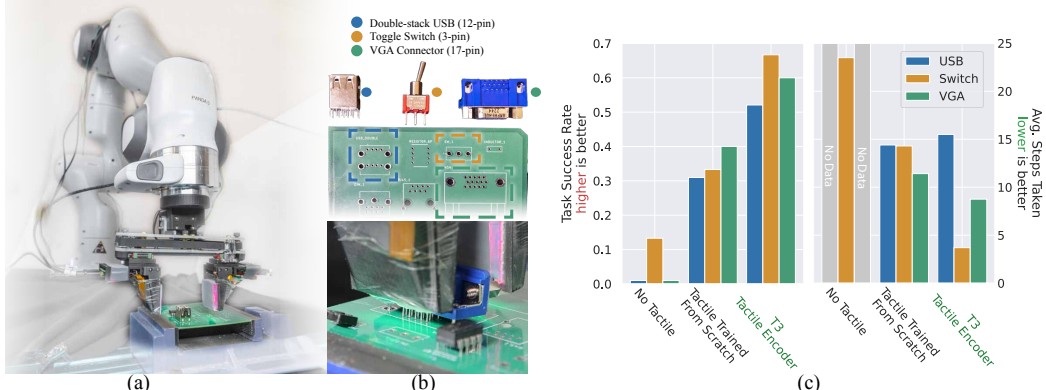

Figure 4: **Real world sub-millimeter robotic insertion tasks.** (a) Hardware setup. (b) The 3 evaluation electronics parts and their respective mounting places on a PCB. (c) Insertion success rate and average steps taken for each task. Policies trained with the T3 tactile encoder achieved the highest success rate and lowest averaged episode length.

the part-in-hand pose and gripper starting pose before each insertion, both generated from a 3D zero-mean, $1mm$ standard deviation normal distribution.

We train and evaluate 3 policies: a baseline policy without tactile input, a policy with tactile input encoded by a neural network trained from scratch, and a policy with tactile input encoded by T3. Besides tactile inputs, all 3 policies have access to the same robot state modality encoded by a MLP, and external vision modality encoded by a pre-trained ResNet18. All three policies take observations of the current step as inputs and predict a 3-DoF action which the robot executes at the next time step. At inference time, the robot executes predicted 3-DoF actions in about 2Hz for up to 30 steps. An episode is deemed successful if the robot successfully inserts the component within 30 steps. The success rate and the average steps taken of successful episodes across 15 episodes are reported in Fig. 4.c.

The results show that (1) the tactile modality is vital for this electronics insertion task, where the vision-only policy failed all tests to insert the two more challenging parts; (2) using a pre-trained T3 as the tactile encoder for this policy helped further improve the overall performance, where the task success rate is higher across all three parts; (3) T3 also helped to reduce the number of tactile exploration steps needed to insert a part.

## 6 Limitations and Future Works

One limitation of the current training pipeline is that the FoTa dataset is unbalanced. Data collected with the 2 most popular sensors constitutes over 50% of the entire dataset. Therefore, the trained policy could also be biased towards those more popular sensors. Another limitation is that the current architecture of T3 focuses on per-image encoding and T3 is trained with explicit labels during pre-training II and fine-tuning. Representation learning on tactile image sequences with sparse or implicit labels can be a future direction of this work. Furthermore, T3 is currently limited to learning representations for camera-based tactile sensors. Extending the FoTa dataset and the T3 framwork to non-camera-based sensors is another future line of work.

## 7 Conclusions

This paper presents T3, a large camera-based tactile sensing framework that transfers across multiple sensors and tasks; and FoTa, a tactile dataset that is by far the largest in quantity and most diverse in sensors and downstream tasks. Experiments show that T3 pre-trained with FoTa improves performance for various tasks, including longer-horizon manipulation tasks such as sub-millimeter electronics insertion, and that T3 transfers to a new sensor-task pairing with little fine-tuning.

**Acknowledgments**

The authors thank Kaiming He for the many insights and discussions around network architecture and training. This paper is partially funded by Toyota Research Institute and Amazon Science Hub. The authors also thank Won Kyung Do from the Stanford ARMLab for providing a DenseTact2.0 for the experiments conducted in the paper.

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

# A  Appendix

## A.1  Visualization of attention maps

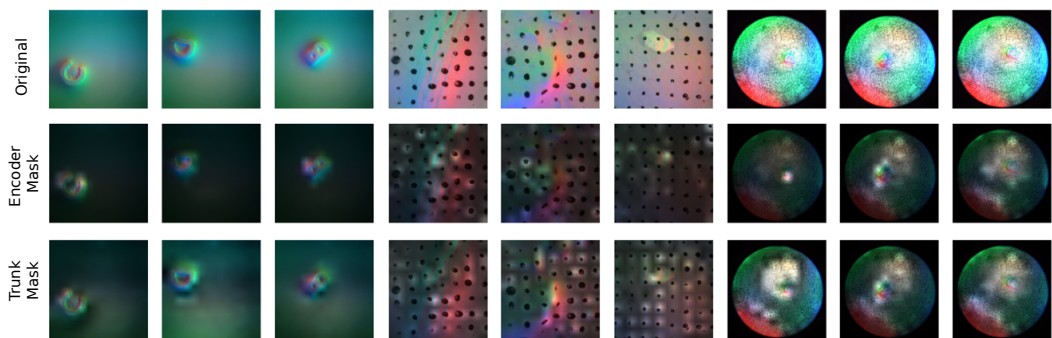

Figure 5: **Visualization of self-attention weights of the encoders and the trunk of pre-trained** `T3`.
Top: the original tactile images. Middle: encoder attention weights applied on the original images as
masks. Bottom: trunk attention weights applied on the original images as masks

We calculate the attention weights $attn_i = softmax(\frac{Q \cdot K}{scale})$ at layer $i$ of each self-attention layers
for all encoders and the trunk. We then obtain a joint attention map, calculated as the product of the
attention weights of all layers within each encoder or trunk, $map = \prod_i attn_i$. This joint attention
map is then normalized and applied on the original tactile images as masks. A few tactile images as
well as them applied with encoder masks and trunk masks are visualized in Fig. 5. Qualitatively, we
observe that the encoder self-attentions highlight the region of the contact object, whereas the trunk
self-attentions are more spread out and they cover other features like the markers on a sensor. This
result seems to show that the encoder is more focused on the sensor agnostic regions, i.e. the "hot"
regions of the contact, while the trunk also attends to sensor specific areas.

## A.2  Public datasets aggregated in the `FoTa` dataset

The `FoTa` dataset does not modify the images or labels from the constituent datasets. Pre-processing
was performed on each of them to encode the tactile images into $.jpg$ format pictures and to extract
all labels into $.json$ format dictionaries. In one of the constituent public datasets, VisGel [26], more
than half of the included tactile images are "flat" images, i.e. tactile images captured when there is no
contact. We sub-sampled VisGel to reduce the amount of flat images. With an image frame $I_i$ and a
true flat frame $I_f$, the variance of Laplacian of the different image $\sigma_i$ is calculated with Eqn. 2. We
then remove all images whose $\sigma < \sigma_t$. We empirically chose $\sigma_t = 4.24$ to remove most flat images.

$$\sigma_i = \sqrt{Var(Laplacian(I_i - I_f))}, i \in \{1...N\} \tag{2}$$

Note that we do not aim to eliminate flat images entirely as flat images also contain important
information such as the sensor geometry and the illumination design. Statistics of the aggregated
datasets in `FoTa` are listed in Tab. 2.

## A.3  Data format

Each constituent dataset of `FoTa` is split into one `train` dataset and one `val` dataset, and each of
them is packaged in a pre-sharded WebDataset [40] format. Each shard of the `FoTa` dataset contains
up to $10,000$ data points packed in one $.tar$ archive, where each data point is composed of one $.jpg$
tactile image and one $.json$ file for various task labels. Utility scripts are also provided to add new
data to the `FoTa` dataset or to modify the pre-processing procedures on the constituent public datasets.

## A.4  Data collection platforms

### A.4.1  7-DoF robotic platform

We built a robotic data collection platform with a 7-DoF Franka Emika robotic arm with a custom-
made parallel jaw gripper attached at the end-effector. Two tactile sensors are attached at the tips

Table 1: Network Sizes

| | $D_{enc}$ | $H_{enc}$ | $L_{enc}$ | $L_{tru}$ | Params incl. all encoders and decoders |
|---|---|---|---|---|---|
| T3-tiny | 192 | 3 | 3 | 9 | 12M |
| T3-small | 384 | 6 | 3 | 9 | 45M |
| T3-medium | 768 | 12 | 3 | 9 | 174M |
| T3-large | 1024 | 8 | 3 | 9 | 308M |

of the parallel jaw gripper, and the transformation between the camera frame and the end-effector frame is accurately measured. During tactile data collection, a test object, which is assumed to be rigid enough to be clamped firmly, is fixed on the workbench of the robot at a known location. The robot explores the object with random movements on all 6 dimensions (translation and rotation), and tactile images from both sensors as well as the robot pose are collected at each step.

### A.4.2 3-DoF gantry platform

Another gantry-based 3-DoF data collection platform was built from a desktop CNC milling machine. The spindle was replaced with a 3D-printed probe, which is attached to the z-axis of the CNC with a 6-axis force torque sensor (MMS101, Mitsumi Electric Company). The probe can be easily swapped out, and we designed in total 6 probes with different textures. During data collection, a tactile sensor is attached firmly to the CNC bed, and the CNC probes the tactile sensor by applying vertical force until the z-axis force reaches a threshold (uniformly sampled from $0.3 - 2.1N$ for most sensors, except GelSight Mini and GelSight Svelte for which we use $1.5 - 5N$ due to higher gel stiffness). Tactile images, probing positions in the sensor's coordinate frame (3D translational poses), and the force values are recorded.

### A.5 Network configurations

In principle, all components of T3 can adopt different architectures, provided they maintain coherence. This section lists the specific architecture and hyper-parameter choices used in this work.

**Encoders** We use VisionTransformer (ViT) [41] blocks as the encoders for tactile images. An image is first resized to $256 \times 256$ then center-cropped to $224 \times 224$, before split into $196$ image blocks each measures $16 \times 16$ pixels. Augmentation is performed during training. Specifically, color jittering is performed for all tasks. Random cropping, horizontal and vertical flipping are performed for all tasks except pose estimation tasks, which rely on the spatial information to make predictions. We use a MLP ratio of $4.0$, an embedding dimension of $D_{enc}$, $H_{enc}$ heads, and $L_{enc}$ layers across encoders for all sensors.

**Trunk** We use ViT blocks as the trunk. We use a MLP ratio of $4.0$, an embedding dimension of $D_{tru} = D_{enc}$, $H_{tru} = H_{enc}$ heads, and $L_{tru}$ layers.

**Decoders** The decoder architecture varies based on the tasks. The reconstruction decoder for masked auto encoding during pre-training I is also a ViT, with a MLP ratio of $4.0$, an embedding dimension of $512$, $16$ heads, and $8$ layers. No pooling is performed after the trunk for this decoder.

The decoders for $\sigma_i$ regression (Eqn. 2) and all classification tasks are MLP decoders. We used the same size of 3 hidden layers $[256, 128, 64]$ for all these MLP decoders. The output dimension varies for the classification tasks depending on the number of categories. Only the classification token is kept from the output of the trunk before it is passed through these decoders.

The decoders for both 3D and 6D pose estimation tasks are constructed with 2 ResNet [32] blocks followed by average pooling then a MLP with hidden layers $[256, 64]$. Before passing through the token outputs from the trunk into the ResNet blocks, we first remove the classification token, then we reshape the output from $[B, T, C]$ to $[B, C, \sqrt{T-1}, \sqrt{T-1}]$. Intuitively, we transform the embedding space back to a 2D map, on which convolution is performed by the ResNet blocks.

Choices of hyper parameters are listed in Tab. 1.

Table 2: Dataset Statistics

| Source | Objects | Labels | Sensor | Marker | Size |
|---|---|---|---|---|---|
| VisGel[26]-small* | Household objects and toys. Flat images from the original dataset were filtered. | Variance of the Laplacian of diff. images | GelSight'17 [12] | Yes | 726,740 |
| TVL [27] | Household objects | semantic object description | DIGIT [44] | No | 82,463 |
| Touch-and-Go [29] | Natural objects | Object IDs | GelSight'17 [12] | Yes | 262,082 |
| Calandra'17 [2] | Household objects | Object IDs, grasp outcome | GelSight'17 [12] | Yes | 24,118 |
| Yuan'18 [4] | Clothes | 11 properties such as smoothness, softness | GelSight'17 [12] | Yes | 494,655 |
| YCB-Sight [28] | Household objects from the YCB object set | Object IDs, 6D poses | GelSight variant No. 1 and simulation | No | 480 real 1,800 sim |
| ObjectFolder-real [30] | Household objects | Object, material IDs | GelSight variant No. 2 | No | 1,417,600 |
| Tippur'23 [18] | Probing sphere | 3D poses | GelSight 360 | No | 13,341 |
| New data collected in-house | Tool objects | Object IDs 6D poses | GelSight Wedge [13] | No | 10,000 |
| | | | GelSight Mini [45] | Yes&No | 24,000 |
| | | | DIGIT [44] | No | 4,000 |
| | Probes with engraved letters | Object IDs 3D poses | GelSight Wedge [13] | No | 3,516 |
| | | | GelSight Finray [21] 2 variants | No | 6,432 |
| | | | GelSight Svelte [19] | No | 5,335 |
| | | | Dense-Tact2.0 [14] | Yes | 3,531 |
| | | | GelSight Mini [45] | Yes | 3,359 |

\* VisGel-small is down-sampled from the original VisGel and most flat images are removed.

