# OpenReview forum: "Transferable Tactile Transformers for Representation Learning Across Diverse Sensors and Tasks"
_robot-learning.org/CoRL/2024/Conference — CoRL 2024_

### Official Review · Reviewer_V6xF · 2024-07-21
**Transformer-based Visuo-tactile representation learning for fast task learning.**

**Originality:** 3
**Technical Quality:** 4
**Clarity Of Presentation:** 4
**Potential Impact:** 3
**Recommendation:** 3
**Confidence:** 5

**Review:**

The paper is clear and easy to follow, addressing the challenge of fast transfer learning of tactile-based policies to new visuo-tactile form factors.

The work used a number of interesting ideas for improved transfer learning across diverse visuo-tactile sensors, including:
- training a common trunk into which features from new sensors can be integrated.
- leveraging both unsupervised learning and supervised learning for training the trunk, encoders and decoders.

However, more qualitative insights are needed to fully understand the results. The paper states that "results show that (1) the tactile modality is vital for this electronics insertion task, where the vision-only policy failed." While the raw success rates may indicate this, it is unclear why solving the task is otherwise impossible. Previous works, such as [1], have solved insertion in a fixed socket setting, so it is a little puzzling that the vision-only policy would completely fail in this setting without even a small success rate. A qualitative analysis of the failure cases would be helpful here.




- [1] Offline Meta-Reinforcement Learning for Industrial Insertion. Zhao et al

**Quality Of The Limitations Section:**

3

**Questions For Rebuttal:**

- The results show that interesting transfer learning occurs when tested on a new sensor-task combination, even though the training data contains the sensor on a different task and the task with a different sensor. I am curious about the performance for a sensor that has not been seen at all in the dataset.

- How wide is the task initialization range (socket position, gripper starting pose, plug in-hand position)? It seems that there is no such randomization. If the setting is fixed, an explanation about the success rate capping at 70% would be helpful. Would training for a longer period or increasing the demonstration data improve the performance? What is the performance bottleneck?

- **Qualitatively**, how does training from scratch compare to T3? Does increasing the dataset size improve performance? A comparative analysis of the dataset size would be valuable to support the paper’s argument that T3 improves sample efficiency.

**Robotics Focus:**

4

**Summary Of Paper:**

This work presents a modular approach to tactile representation learning that is compatible with different visuo-tactile sensors and valuable for multi-task learning. The approach achieves this by curating a large dataset of visuo-tactile sensor recordings across various tasks. Different encoders are used for each sensor and different decoder heads for each task, allowing a common central backbone to be learned. This backbone enables various encoder/decoder combinations to be plugged in, depending on the sensor/task combination. The model is trained using unsupervised learning loss in conjunction with task-specific supervision loss. Results suggest that using the pretrained central backbone accelerates learning with new sensors or on new tasks.

**Summary Of Recommendation:**

Overall, the paper is well-written, scoped, and targeted towards improving tactile representation learning for few-shot transfer to new visuo-tactile sensors. Additional analysis on the sample efficiency and qualitative insight will further strengthen the paper.

---

### Official Review · Reviewer_aci4 · 2024-07-21
**Tactile representation learning with great potential**

**Originality:** 3
**Technical Quality:** 2
**Clarity Of Presentation:** 3
**Potential Impact:** 3
**Recommendation:** 2
**Confidence:** 4

**Review:**

The quality of the paper is good. The work is written in clear language, and technical details are described and depicted clearly in the main text and figures. The reviewer appreciates the authors’ efforts in pulling up such a comprehensive dataset for tactile sensing. The main strength of the paper is that it gives a good review of the existing vision-based tactile sensing in terms of training dataset and various tasks. The paper shows the potential of learning a shared representation of tactile sensing across diverse sensors and tasks.

Nonetheless, the reviewer has some concerns regarding the originality and significance of the paper to a broader audience in the relevant field. The main weaknesses of the paper are:
1. While T3 achieves zero-shot transferability and performs well with minimal fine-tuning across new sensors and tasks, the diversity of tactile sensors is largely limited to GelSight or its variations. These sensors share common sensing principles. The authors are suggested to discuss this in the limitations and future works.
2. Some of the conclusions are not convincing with quantitative solid evidence. Refer to the list of issues below.
3. The results is only in terms of numbers in performance. More investigations and understanding are expected.

**Quality Of The Limitations Section:**

2

**Questions For Rebuttal:**

1. As shown in Figure 1, the architecture of T3 includes sensor-specific encoders and task-specific decoders to work with any sensor-task combination. This means we need to create a new encoder every time we include a new sensor. To what extent can we call it a new sensor? Is it possible that those similar sensors share the same encoder? It seems to the reviewer that this architecture limits the generality of the framework to novel sensors. Did the authors investigate the benefits of having an encoder for each sensor?
2. Some results are over-stated. For example, line 212 says, “We observed that pre-training significantly improved the evaluation performance with all network configurations with an average improvement of 31%”. However, as shown in Fig. 3a, finetuning from pre-train 1 seems only slightly better than training from scratch for medium model size. The significant improvement with a large model size may simply be due to the mismatch between the model size and the amount of training data. (The data size is not mentioned in Section 5/1).
3. The comparison in 5.1 may not be fair. It is a common understanding that transfer learning is more effective than training from scratch. The experiment setup does not convince whether the pretraining has successfully learned the tactile representation. What if the authors use pre-trained weights from other datasets?
4. The authors should investigate the learned tactile representation more instead of simply showing the different numbers in performance. Are there any more intuitive or fundamental understandings? Is there any pattern in the shared trunk?
5. The zero-shot transferability is not towards totally new sensors or tasks but their combinations. As shown in Table 1 in the Appendix, the sensor Gelsight Svelte has only two tasks: classification and 3D poses. Does that mean the pre-trained model has never seen any data on Gelsight Svelte?
6. The dataset is said to include 11 tasks. However, as shown in Fig. 2, most task labels are actually the same type of task, i.e., object classification. Unlike tasks such as force estimation or slip detection, the object classification task is much more affected by the tested objects than the sensors. So, a natural question is: Are the tasks included in this paper the right choices for learning tactile representation?

**Robotics Focus:**

4

**Summary Of Paper:**

The paper proposed a framework T3 for tactile representation learning that scales across multi-sensors and multi-tasks. The framework captures the shared latent information across different sensor-task pairings by constructing a shared trunk transformer with sensor-specific encoders and task-specific decoders. This paper also presents a novel Foundation Tactile dataset over 3 million data points. T3 pre-trained with FoTa achieved a certain degree of zero-shot transferability and good small-data-finetuning transferability. The creation and open-sourcing of the FoTa dataset represent a substantial contribution to the field, providing a valuable resource for future research.

**Summary Of Recommendation:**

Overall, this work is a valuable addition to the literature and holds promise for future advancements in tactile sensing. However, further investigations are required to conclude convincing results.

---

### Official Review · Reviewer_1X3T · 2024-07-23
**good contribution to the camera-based tactile sensing research community**

**Originality:** 3
**Technical Quality:** 4
**Clarity Of Presentation:** 5
**Potential Impact:** 3
**Recommendation:** 3
**Confidence:** 4

**Review:**

### Strengths
- Collecting many existing tactile sensing dataset, adding new self-collected data, and making them available in a single dataset is a good contribution to the tactile sensing research community.
- The foundation model training uses a novel two-step pre-training method to improve data efficiency, and the efficacy of both the steps is experimentally verified.
- Section 5.3 shows that this pretrained model is indeed useful as a starting point for unseen manipulation tasks that require a tactile sensing component.
- The paper is well written and easy to understand.

### Weaknesses
- A pre-trained foundation model is most useful when training from scratch for a task is difficult, for example because of a scarcity of labelled training data for that task. This paper evaluates that the proposed model is indeed useful in this kind of scenario (Fig. 3a) but the evidence is weak. The difference in accuracy between a model trained from scratch vs. finetuned not very large. Additionally, this experiment tests only one task (object classification). It is not clear from the text (please clarify) but the foundation model is also trained on this task.
- some possibly typing errors:
  - L165 processes -> possesses
  - L216 base -> medium

**Quality Of The Limitations Section:**

3

**Questions For Rebuttal:**

Referring to the weakness mentioned above, I suggest the authors to do unseen task experiments, like the ones in Section 5.2, and compare the performance of a model trained from scratch on the unseen task to the performance of the finetuned model. This will shine a light on the usefulness of the proposed foundation model in a practically relevant setting.

**Robotics Focus:**

4

**Summary Of Paper:**

Firstly, this paper proposes a large scale camera-based tactile sensing dataset with > 3M data points. Secondly, a transformer model is trained on this dataset and experiments are conducted to verify that the model is useful as a pre-trained starting point for various unseen tasks.

**Summary Of Recommendation:**

I am recommending to accept this paper because of its strengths and the usefulness of the contribution to the research community. The paper can be improved by addressing the weakness mentioned above.

---

### Author Rebuttal · Authors · 2024-08-08

We thank all reviewers for their insightful comments and constructive reviews. Overall, we appreciate that all reviewers acknowledged this paper’s contributions. On the weakness’ side, most commonly raised concerns include more detailed benchmarking of the pre-trained networks, as well as insights and qualitative analysis of the learned representations. To answer those questions, we have ***added two experiments (fine-tuning with different amounts of data, visualization of attention maps)***, and overhauled the experiment and discussion section of the paper. We have uploaded a revised manuscript and all changes in the pdf are marked in green.

In summary:

**An additional fine-tuning sample efficiency experiment**
We added an additional sample efficiency experiment in Sec 5.1 where we also tested the fine-tuning performance against different amounts of fine-tuning data. In summary, our new experiment shows that pre-trained models and larger models both perform better than their counterparts with a smaller pool of pre-training data. This proves that pre-trained T3 enables **faster transfer learning**. We want to especially thank reviewer V6xF for recommending this new experiment.

**Qualitative analysis of the learned representations**
We added visualizations of the encoder and trunk attention maps in Appendix A.1. Qualitatively, we observe that the encoder self-attentions highlight the region of the contact object, whereas the trunk self-attentions are more spread out and they cover other features like the markers or profiles of a sensor. More details can be found in A.1.

**Clarification on the zero-shot transferability experiment with novel sensors**
The zero-shot transferability experiment of Sec 5.2 was conducted with novel sensors and seen tasks. The novel sensors (GelSight Svelte, DenseTact2.0) were sensors that the model had not seen during pre-training. The clarification and pose regression tasks were pre-trained with GS Wedge, GS Finray, GS Mini. When running the zero-shot transferability experiment, we use ec-wedge as the encoder for GS Svelte, and ec-mini for DenseTact (due to similarities in the construction of the paired sensors). Then we attach dc-cls and dc-reg to obtain the inference performance. We have updated the text of Sec 5.2 to include this clarification.

We thank the reviewers again for their time and thoughtful reviews. We hope our new experiments and clarifications will address the concerns raised by the reviewers.

---

### Decision · Program_Chairs · 2024-09-04

**Decision:**

Accept

**Comment:**

This paper addresses learning tactile representation across a variety of vision-based tactile sensors and tasks.

The reviewers agree in the relevance and importance of this task and highlight several strengths including the dataset, the novelty in the training paradigm, and the overall clarity of the presentation.

Reviews, however, highlight a number of weaknesses with the strength and presentation of the results. Most importantly the reviewers note that there is only a marginal increase in performance using pre-training and there's a lack of broader insights / qualitative analysis provided in the results.

The authors addressed the reviewers concerns sufficiently well in the rebuttal phase, that I can now recommend acceptance.